# Storm Surge Barrier Protection in an Era of Accelerating Sea-Level Rise: Quantifying Closure Frequency, Duration and Trapped River Flooding

**Ziyu Chen** [1,*], **Philip Orton** [1,*] **and Thomas Wahl** [2]

[1] Davidson Laboratory, Stevens Institute of Technology, Hoboken, NJ 07030, USA

[2] Department of Civil, Environmental and Construction Engineering, University of Central Florida, Orlando, FL 32816, USA; t.wahl@ucf.edu

\* Correspondence: zchen44@stevens.edu (Z.C.); porton@stevens.edu (P.O.);
Tel.: +1-(973)-393-6846 (Z.C.); +1-(201)-216-8095 (P.O.)

**Abstract:** Gated storm surge barriers are being studied by the United States Army Corps of Engineers (USACE) for coastal storm risk management for the New York City metropolitan area. Surge barrier gates are only closed when storm tides exceeding a specific "trigger" water level might occur in a storm. Gate closure frequency and duration both strongly influence the physical and environmental effects on enclosed estuaries. In this paper, we use historical observations to represent future storm tide hazard, and we superimpose local relative sea-level rise (SLR) to study the potential future changes to closure frequency and duration. We account for the effects of forecast uncertainty on closures, using a relationship between past storm surge and forecast uncertainty from an operational ensemble forecast system. A concern during a storm surge is that closed gates will trap river streamflow and could cause a new problem with trapped river water flooding. Similarly, we evaluate this possibility using historical data to represent river flood hazard, complemented by hydrodynamic model simulations to capture how waters rise when a hypothetical barrier is closed. The results show that SLR causes an exponential increase of the gate closure frequency, a lengthening of the closure duration, and a rising probability of trapped river water flooding. The USACE has proposed to prevent these SLR-driven increases by periodically raising the trigger water level (e.g., to match a prescribed storm return period). However, this alternative management approach for dealing with SLR requires waterfront seawalls to be raised at a high, and ongoing, additional future expense. For seawalls, costs and benefits will likely need to be weighed on a neighborhood-by-neighborhood basis, and in some cases retreat or other non-structural options may be preferable.

**Keywords:** storm surge barrier; sea level rise; hazard assessment; risk reduction; adaptation; flood duration; hydrodynamic model; New York City

---

## 1. Introduction

Coastal cities around the world are exploring structural engineering options for defending against extreme storms and the resulting surges of ocean water that can cause massive flooding. Storm surge barriers or tide gates can effectively protect harbors and minimize flooding, property damage, and loss of life during large storms.

Surge barrier systems costing tens of billions of dollars are being evaluated by the United States Army Corps of Engineers (USACE) as one of a number of options for flood risk reduction for the New York City metropolitan area [1]. The USACE estimates that the cost of coastal flood risk is very high in the region, at $5.1 billion year$^{-1}$ in 2030 and $13.7 billion year$^{-1}$ at 2100 (under an intermediate sea level rise trajectory; [1] p. 62). The decision of whether or not to build surge barriers to protect one of

the nation's main commercial hubs and ports, crossing one of the world's most iconic estuaries, is a major decision worthy of thorough analysis of potential impacts.

These barriers typically span the opening to a harbor or river mouth and include gates that are only closed when storm surges are expected. When gates are open, any fixed infrastructure that reduces the flow cross-sectional area of an inlet or estuary channel leads to some degree of continuous physical changes throughout the estuary [2,3]. Closure of the barrier gates has a more acute effect, reducing water flow and tidal exchange, which in turn could affect water quality and ecological processes [4–6]. Consequently, gate closure frequency and duration are major determinants of surge barrier estuary effects. The present-day and future gate closure frequency and duration are important environmental impact indicators to understand the impacts of the storm surge barriers.

Sea-level rise (SLR) is a major factor that can lead to increased closure requests, raising management challenges because SLR will lead to the increase of water level exceedances above coastal flooding thresholds. For example, Figure 1 shows that the present-day exceedance probability above the "moderate" National Weather Service (NWS) flood threshold at the Battery tide gauge location (National Oceanic and Atmospheric Administration (NOAA) tide gauge station 8518750 at southern Manhattan, NYC) is very small, but it will become much larger with 60 cm SLR (50th percentile, Representative Concentration Pathway (RCP) 4.5) later this century [7]. The barrier closure duration (number of successive tidal cycles where water level exceeds a flood threshold) may also be lengthened with future SLR, as demonstrated in Figure 2. Those changes would intensify any environmental impacts on the estuary.

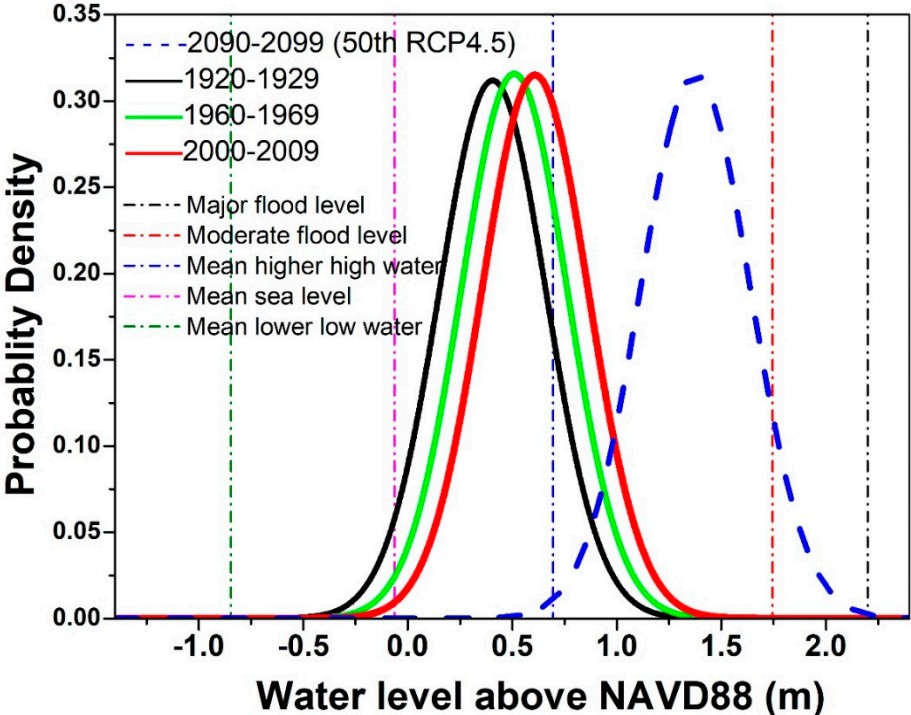

**Figure 1.** The probability of semidiurnal water level peaks exceeding National Weather Service flood thresholds (moderate, major) increases with sea-level rise (SLR). Probability density functions shown here are based on historical water level data from the Battery, New York City, and with 50th percentile regional SLR projections to 2090–2099 for a moderate emissions pathway (RCP 4.5).

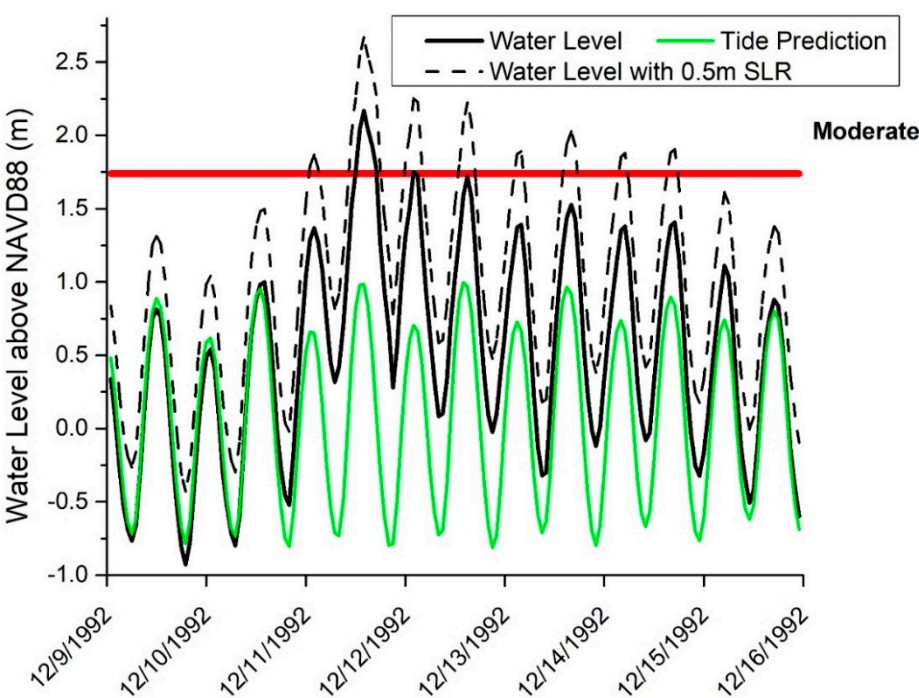

**Figure 2.** Example of a flood event (1992 Nor'easter) with 3 consecutive semidiurnal peak water levels above the "moderate flood" threshold (red line). Dashed lines show the water level change with 0.5 m SLR, leading to a closure duration change from 3 to 8 tidal cycles. This is an example that requires a long-duration gate closure, over multiple semidiurnal periods.

A lengthened barrier closure duration will also increase the trapped river flood risk behind the barrier. Recent studies have shown that rain (or streamflow) and coastal surge are correlated at many coastal cities around the world, and evidence at some locations that their correlation is increasing, including for New York City (NYC) [8,9]. Climate change could exacerbate future rain intensity and streamflow [10], rendering it even more important to assess flood risk from the co-occurring trapped streamflow due to the closed storm surge barrier.

Prior studies have investigated the continuous effects of open barriers on physical estuary conditions [2,3,11], but there has been a limited number of academic assessments focused on surge barrier closure frequency, duration, and trapped river flooding. These factors also require careful consideration for any proposed surge barrier system. Kirshen et al. [12] evaluated how the gate closure frequency would increase in the future for a hypothetical Boston surge barrier, and performed a preliminary multi-disciplinary assessment of surge barrier effects. However, they did not address how the duration of individual closures will evolve. Also, they did not integrate the forecast effect on the gate closure frequency. To guarantee that the gate closure frequency is less than an acceptable maximum, it is important to introduce a margin of error regarding the effects of inaccuracy of water level forecasting on the calculation of the frequency of exceedance. The forecast uncertainty is not only proposed by the USACE as the required surge barrier design criteria ([1], p. 37 in Engineering Appendix), but also a significant factor that has been considered and addressed in determining surge barrier design from prior Dutch experience [13].

The goals of this paper are:

(1) to demonstrate a simplified, transferable framework for barrier closure analyses that builds upon the work of Kirshen et al. [12];

(2) to estimate the surge barrier gate closure frequency and duration and their future evolution with SLR; and

(3) to assess the probability of trapped river water flooding during periods of gate closure.

Sections below detail the methods, results, discussion and primary conclusions of the research, and Supplementary Materials include additional figures, data and codes for reproducing the primary analyses.

## 2. Methods

The general methodological approach of this research is to use historical tide gauge data to represent empirically stochastic storm-driven and tide-driven variability in harbor water levels, superimpose SLR on these water levels, and compute the frequency and duration of exceedances (and thus, required gate closures) for each year going forward. We also create a model of 24 h forecast uncertainty for incorporation into the water level data. Then, we quantify the trapped water elevations at the NYC for all the present/future "gate closure triggered" events. Figure 3 shows the flow chart regarding the detailed procedures and steps for the analysis. In this research, we make several simplifications on the dynamic processes of water level change as well as the closure operation of the barrier system; these are further explained and discussed in the following subsections.

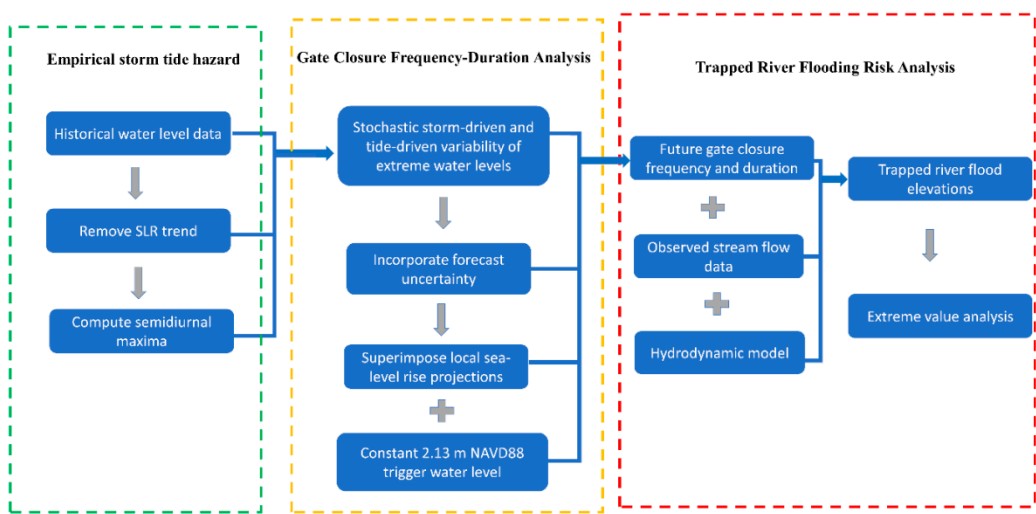

**Figure 3.** Diagrammatic representation of the processes and analyses in "Methods" section.

### 2.1. Gate Closure Trigger Water Level

Two surge barrier closure management regimes are possible—one is based on a constant water level threshold in forecasts that is a "trigger" for gate closure (e.g., [14]), and the other is a constant annual exceedance probability (AEP) (or inversely, return period), in which case the water level trigger is updated periodically as sea levels rise. Our analyses focus mainly on the first management regime, given that many of the world's existing surge barrier systems are being closed more frequently due to SLR (e.g., the English Thames barrier [15]; the Dutch Rijkswaterstaat and Maeslantkering barriers [16]). However, we will also examine how our analyses inform the second management regime, and discuss the two regimes in Section 4.

We focus mainly on a constant water level trigger of 2.13 m (7 feet) above the North American Vertical Datum of 1988 (NAVD88) in our analyses. This is just below the National Weather Service (NWS) "major flood level" at the Battery, New York City (2.20 m NAVD88), indicating extensive inundation and significant threats to life and property. The value of 2.13 m is the most recent value being used by the USACE in their assessments (B. Wisemiller, USACE New York District, pers. comm, January 2020), balancing protection with environmental concerns by keeping the closure frequency below 0.5 year$^{-1}$ for several decades into the future. The USACE has also proposed that if sea levels rise sufficiently for closure frequencies to exceed 0.5 year$^{-1}$, they will then use a constant AEP trigger ([1], p. 69).

## 2.2. Empirical Storm Tide Hazard

The observed water level data is utilized to create the stochastic storm-driven and tide-driven variability in harbor water levels, hereafter referred to as the storm tide hazard. Here, we ignore any changes to tides or storm surge caused by the fixed infrastructure of barriers. Based on past research [3], various hydrodynamic models' results indicate that changes to tidal range conveyance can be kept small, if this location's surge barriers have an open gated flow area that is greater than about 50%, which is the case for all the barrier systems being considered by the USACE ([1], pp. 26–27 in Engineering Appendix). The historical observed hourly water level data from 1920 to 2019 at the Battery tide gauge location is used for this study. We remove a cubic fit to annual mean sea level from 1920 to 2019 to eliminate the effects of long-term SLR but retain intra- and inter-annual variability. Then, we create a semidiurnal maxima dataset from the detrended hourly water level data to represent the stochastic storm-driven and tide-driven signal in harbor water levels. To characterize future flood hazards, we are only considering future SLR and not considering future storm tide climatology changes in this study. Based on recent studies [17,18], projected changes to storm tide climatology in New York City will contribute a relatively minimal (below 10% relative) impact on changing extreme water levels compared with SLR by the end of the 21st century.

## 2.3. Impacts of Forecast Uncertainty

In the practical operation of the surge barrier system, decisions on gate closure will be based on forecasts of future water levels, which will always have uncertainty. Notably, the forecast water level uncertainty is usually larger for more extreme events, such as deep extratropical low-pressure systems or hurricanes (e.g., [19,20]). The forecast water level needs to come 24 h or even earlier in advance of the projected flooding event giving time for the cumbersome gate closure operation before the event hits. Typically, the surge barrier manager will refer to the spread of an ensemble of forecast water levels and close the barrier even if there is a small chance of the trigger threshold being exceeded. This will inevitably cause more closures, but minimize the risk (monetary and political) of not closing the barrier and having a surprise, damaging flood (a false negative).

We quantify typical uncertainty in water-level forecasts and its dependence on storm surge by utilizing the past 4 years of forecast results from an operational forecast system [19,21]. The 95th percentile forecast is chosen as a hypothetical value for barrier closure decisions, which leads to a probability of 5% of false negatives. A "high-end uncertainty" (95th percentile—minus the 50th percentile at the time of the forecast peak) is defined as an additional increment to add to historical observations to represent the average effect of forecast uncertainty.

The storm surge component is the key factor that influences the forecast uncertainty, due to uncertainty in the meteorological forecasts. We investigate the relationship between storm surge height and forecast uncertainty by using water-level observations and past ensemble forecasts. First, we perform a harmonic analysis (considering 37 harmonic constituents) to extract the tide and remove it from the observed water level to obtain the surge values. Then, we match the surge values for each water-level peak with the corresponding ensemble forecast. We capture the semidiurnal maxima from the observed water levels, and find the historical ensemble forecast that was delivered 24 h prior to arrival of that peak and compute the "high-end uncertainty". We selected only the top 20 high water-level events (a threshold being exceeded with a 5 year$^{-1}$ frequency) from the dataset and use these as samples to fit the regression model. In this way, we obtained a relationship between high-end uncertainty and surge (Figure S1 in Supplementary Materials; uncertainty = 0.214 × surge + 0.100; Coefficient of determination $r^2$ = 0.523) and add it to observed maxima. While the formula was derived during a period with no extreme events, we also find it to give a reasonable result for a retrospective forecast of Hurricane Sandy (2012). When we use this relationship to compute the "high-end uncertainty" for Hurricane Sandy, the result is 0.66 m. This is close to the corresponding 0.5 m high-end uncertainty published before using the same ensemble forecast model [21].

### 2.4. Local Sea-Level Rise (SLR) Projections

Probabilistic sea level rise scenarios are considered to quantify gate closure frequency and uncertainty in the future. We utilized relative SLR projections [22,23] for the Battery under two emissions scenarios (RCP4.5 Intergovernmental Panel on Climate Change (IPCC) moderate emissions pathway; RCP8.5 IPCC high emissions pathway). These SLR projection studies provided 20,000 Monte Carlo samples at decadal time steps. After unifying the water-level data and SLR data to the same datum, we compute the 10th, 50th and 90th percentile of SLR and linearly superimpose them on the synthesized peak water level maxima (those with uncertainty added, as described above) to represent their future evolution with future SLR and calculate the surge barrier closure frequency and duration. Here, we do not consider the non-linear interaction between storm tides and SLR, we simply superimpose the SLR data to the water level to represent the future water level under SLR. This static assumption has been tested and verified at this area through past research using hydrodynamic modeling [24,25]; model results are very close to the superposition of water level and SLR. We also utilize the USACE intermediate/high SLR data [26] for analysis to compare with our results.

### 2.5. Gate Closure Frequency and Duration Analysis

In quantifying the future gate closure frequency and duration, we are considering the actual closure operations; if there are temporally-consecutive water level peaks (i.e., high tides) above the threshold, it will be counted as one gate closure. This assumes the gate will not be closed and opened within one tidal cycle. Thus, the "duration" is defined as the number of semidiurnal tidal cycles during which the threshold is exceeded (see example in Figure 2). This is reasonable because the processes of opening and closing gates of surge barriers both typically take several hours [12]. The gate will stay closed in a long-duration flood event (e.g., extra-tropical storm), as has been assumed in other surge barrier closure modeling studies (e.g., [27]).

We ignore spatial variations in water level, using only water levels from the Battery in our analysis. This is a reasonable way of creating a simplified analysis and avoiding a large quantity of hydrodynamic model simulations, because spatial variations in water levels around New York and New Jersey Harbor are typically small (on the order of 10% of the storm tide; e.g., [28]).

### 2.6. Trapped River Flooding Analysis

A combination of hydrodynamic modeling of water levels during six past storms (Table 1) with three barriers closed (Verrazano Narrows barrier; Throgs Neck barrier; Arthur Kill barrier) and 25 years of historical observed streamflow data are used to quantify the annual frequency of trapped river water flooding during gate closures. Here, we are only looking at the trapped water rise at the Battery location in the model run, as NYC is the most populated location behind the barrier. Considering that there are small spatial variations along the Hudson River normally (averaging only a ~40 cm rise over the 225 km tidal reach; [3]) and these will be reduced due to the gate closure at the ocean end, we do not examine the spatial variations. The streamflow observations are used to obtain accumulated water volume during closure events and modelling is used to turn this into the water elevation increment at the Battery and infer a relationship between the two based on the historical storm scenarios. First, we use the historical hourly streamflow data from 16 available the United States Geological Survey (USGS) gauges in the study area to construct the total streamflow volume input time series from 1990–2014. Figure S4 (in Supplementary Materials) shows the study area with all the USGS gauge locations, the three barriers' closure system and the Battery location. There are gaps of several gauges that do not have the streamflow records for some periods of time. These gaps are closed by using nearby available gauge data and scaling by relative watershed drainage area. Then, we utilize the gate closure duration results (assuming constant 2.13 m trigger water level) to calculate the simultaneous total trapped water volume for each gate closure triggering event.

**Table 1.** Six historical storms modeled with surge barriers closed, for trapped water analysis.

| Simulation Dates | Storm Name | Storm Type | Peak Storm Tide (m, NAVD88) | Peak Streamflow Total (m³/s) |
|---|---|---|---|---|
| 17 January 1996–23 January 1996- | None | Extratropical cyclone | 1.14 | 6458.3 |
| 6 January 1998–12 January 1998 | 1998 Ice Storm | Extratropical cyclone | 0.99 | 4713.9 |
| 13 September 1999–19 September 1999 | Floyd | Tropical cyclone | 1.11 | 2994.4 |
| 31 March 2005–5 April 2005 | None | Wet extratropical cyclone | 1.23 | 6566.6 |
| 9 April 2007–19 April 2007 | Tax Day storm | Extratropical cyclone | 1.53 | 6250.0 |
| 22 August 2011–1 September 2011 | Irene | Tropical cyclone | 1.97 | 7897.2 |

To illustrate how trapped water flooding changes with SLR, we superimpose three SLR scenarios (0.3 m, 0.6 m and 1 m SLR) to the storm tide maxima over the same period, to evaluate cases where the surge barrier closure would be triggered. We then calculate the trapped water volume based on two possible gate operation methods: (1) the gates close at the slack tide at the Battery and reopen at the slack tide, and we assume streamflow starts to accumulate from mean sea level at the Battery for each triggering event; and (2) the gates close at low tide and reopen at low tide, in which case we assume streamflow starts to accumulate from mean low water level at the Battery. This mean low water value is raised with superposition for the sea level rise scenarios.

The New York Harbor Observing and Prediction System (NYHOPS) hydrodynamic model is used to model six historical storms [29] in order to obtain an empirical relationship between trapped water volume from the observed tributaries and water level rise elevation at the Battery. The model accounts for freshwater inputs from all sources in this area [28,30] and includes the three barriers' closure system. We used the relationship obtained from the hydrodynamic model to transfer the trapped water volume to water level increase for all gate closure events. Figure S5 (in Supplementary Materials) shows the empirical relationship between observed trapped water volume coming from different tributaries and modeled rise in water level elevation at the Battery. Multiple types of historical storm (Table 1) are used as the model inputs for constructing the relationship.

Lastly, we perform extreme value analysis on the trapped water elevation data for the different future SLR scenarios to estimate the probability of surpassing the 2.13 m flood threshold. We fit the generalized Pareto distribution to the trapped river flood elevation data and calculate the exceedance probability for the 2.13 m NAVD88 flood threshold for each case. To best model the distribution's tail, we choose the top 25 trapped river flood elevations (a 1 year$^{-1}$ closure frequency based on the 1990-2014 data) as samples in each SLR scenario. For sea-level rise scenarios where the number of gate closures is smaller than 25 times (e.g., 8 closures in 25 years, for the 0.3 m scenario), we do not fit a distribution for two reasons: (1) there are too few points to robustly fit a distribution and (2) the probability is effectively zero, because none of the trapped water levels is within 1 m of 2.13 m.

## 3. Results

### 3.1. Gate Closure Frequency Analysis

Analysis of barrier gate closures contrasted two potential approaches for managing barrier gate closure: (1) the case of a constant water-level trigger and (2) the case of a constant AEP such that the trigger water level is raised periodically as sea levels rise. Figures 4–6 demonstrate how SLR could affect barrier closures based on a constant water level trigger. Figure 4 contrasts the influence of the

trigger water level (moderate flood vs. 2.13 m flood level) on the number of closures, which shows that the trigger water level has a strong influence on the number of closures. A 0.39 m rise of trigger water level (1.74 m NAVD88 to 2.13 m NAVD88) makes about a factor of 10 difference of the closure frequency.

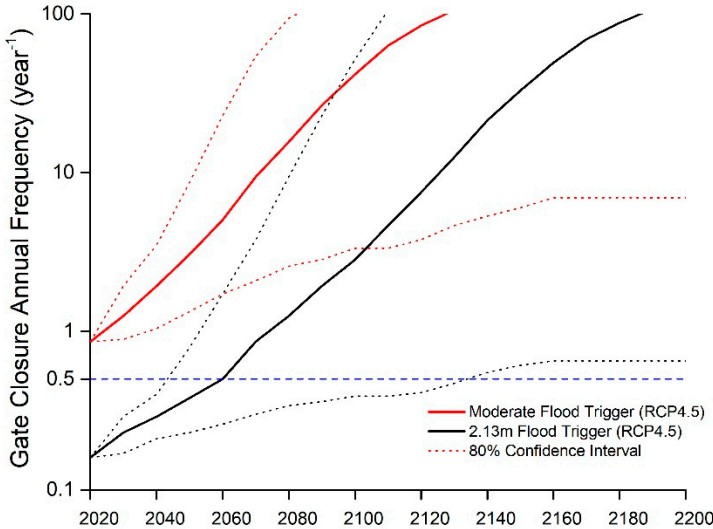

**Figure 4.** Gate closure frequency analysis results based on constant water level triggers and RCP 4.5 sea level rise; red lines are using moderate flood trigger (1.74 m NAVD88), black lines are using 2.13 m flood trigger. The horizontal dashed line is the given maximum closure frequency (0.5 year$^{-1}$). The y-axis is cut off at 100 year$^{-1}$ since the closure frequency above 100 year$^{-1}$ is far beyond our research interest.

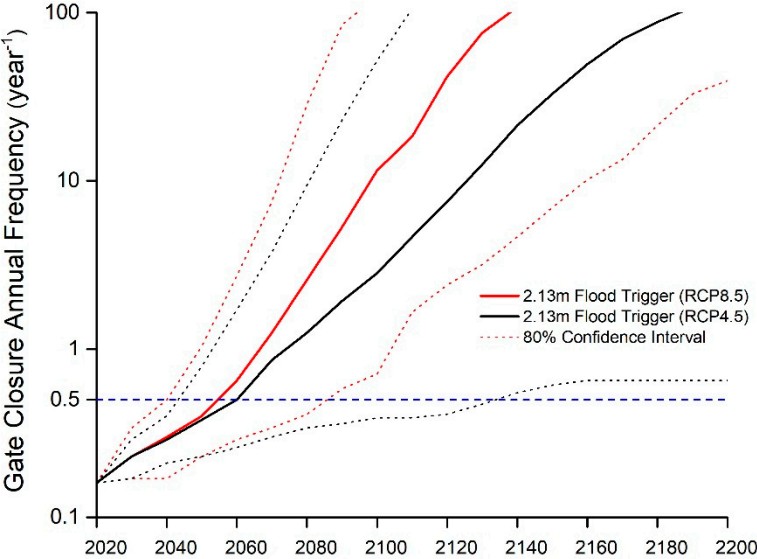

**Figure 5.** Gate closure frequency analysis results based on constant 2.13 m flood trigger; red lines are using the RCP 8.5 sea level rise; black lines use the RCP 4.5 sea level rise. Dashed line is as Figure 4.

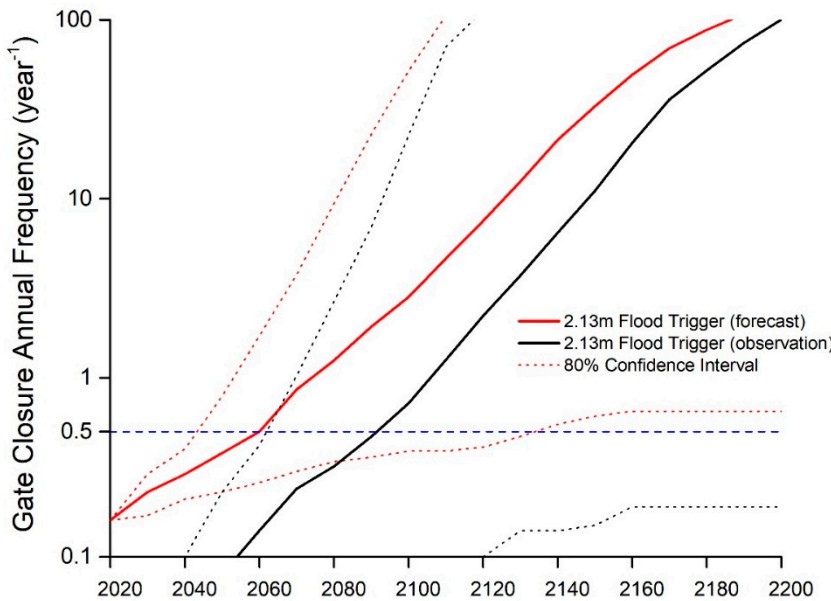

**Figure 6.** Gate closure frequency analysis results based on a constant 2.13 m flood trigger and RCP 4.5 sea level rise; red lines are based on an estimated storm tide hazard with forecast uncertainty (the 95th percentile); black lines are based on the storm tide hazard with zero forecast uncertainty. Horizontal dashed line is as Figure 4.

Figure 5 compares different SLR scenarios (RCP 4.5 vs RCP 8.5) effects on the gate closure frequency; The sensitivity of the number of closures to these two greenhouse gas emissions trajectories becomes larger with time, However, the difference of arrival time of the 0.5 year$^{-1}$ maximum closure frequency between these two is small (2054 versus 2060).

The forecast uncertainty has a large effect on gate closure frequency, particularly at or below the 0.5 year$^{-1}$ frequency threshold). Figure 6 shows the effect of incorporating the forecast uncertainty on the number of annual gate closures (comparing red and black lines). For example, in 2060 the central estimate of the closure frequency by using synthesized forecast water level is reaching the 0.5 year$^{-1}$ frequency threshold, while the corresponding closure frequency calculated without forecast uncertainty ("observation") is only about 0.15 year$^{-1}$. After considering the forecast uncertainty, the threshold of 0.5 year$^{-1}$ is reached 30 years earlier, in 2060 instead of 2090.

The alternative gate closure management regime of a constant gate closure AEP requires a rise in the trigger water level in future decades (Figure 7). As the gates would not be closed for lower water levels, this would require a higher elevation for shoreline protection. The closure AEP was assumed to be constant at 0.5 year $^{-1}$. The results indicate the useful time horizon of the barrier/seawall combined system based on RCP 4.5/ RCP 8.5 future greenhouse gas emissions trajectories (2020–2200) and USACE SLR projection (2020–2100, [26]). For a risk-averse planning perspective, we show high-end 90th percentile scenarios (dash lines) with the 50th percentiles (solid lines), but we omit the 10th percentiles. The 90th percentile results show that using the 2.13 m flood level as the trigger can make this system functional at least until 2040; raising the trigger water level to 2.6 m extends functionality at least until 2070.

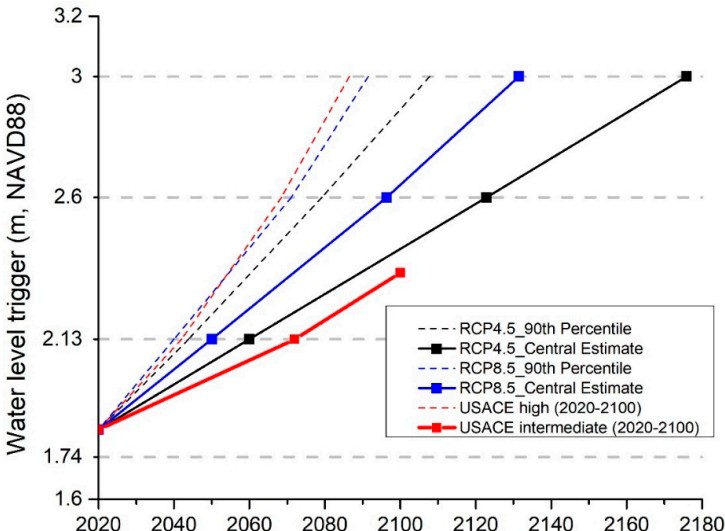

**Figure 7.** Future evolution of the water-level trigger based on a constant AEP (0.5 year$^{-1}$) and three sea level projections: RCP 4.5 (black), RCP 8.5 (blue) and the USACE intermediate projection (red). These results use observed data incorporated with forecast uncertainty (the 95th percentile).

### 3.2. Gate Closure Frequency-Duration Analysis

While gate closure frequency rises exponentially with SLR, the gate closure duration rises more slowly. For each flood event, we calculated the corresponding gate closure duration based on its flood duration time (See example from Figure 2). Here, we obtain the result of the closure frequency for different closure duration events. Figure 8 shows the exceedance frequency (frequency for the flood events equal or greater than N semidiurnal tide cycles) for flood events with various durations (ranging from 1 semidiurnal tide cycle to 10 semidiurnal tide cycles). We assume a constant 2.13 m flood water level trigger and RCP 4.5 future greenhouse gas emissions trajectories for the analysis. The median (50th percentile) scenarios are shown for 2050 and 2100 (solid lines). For a risk-averse planning perspective, we also show high-end 90th percentile scenarios (dash lines), but we omit the 10th percentiles.

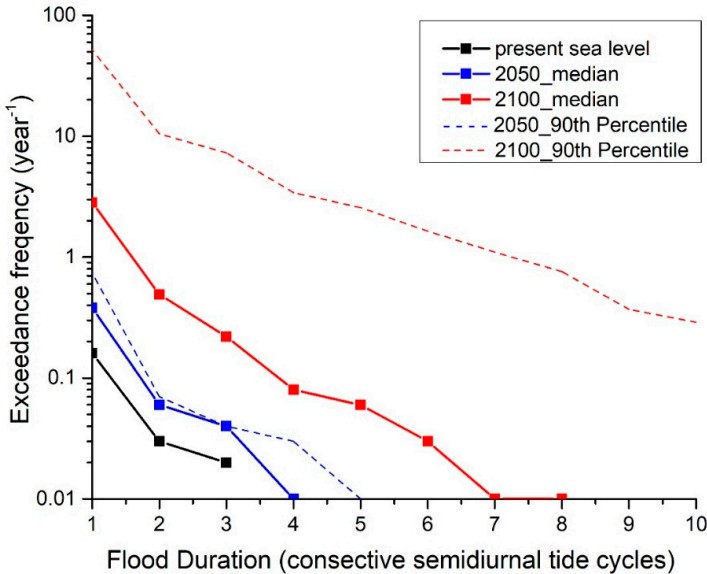

**Figure 8.** Gate closure frequency-duration analysis results based on constant 2.13 m flood trigger and RCP 4.5 sea level rise; red lines are using the SLR projection at 2100; blue lines are using the SLR projection at 2050; black line is using present day sea level.

### 3.3. Trapped River Water and Flooding

Figure 9 demonstrates the potential risk of the trapped river water flood due to closures of the surge barrier system. As sea level rises, the combination of a higher water level starting point, and increased closure frequency and duration leads to increased trapped river water and a higher probability of trapped river water flooding. With the constant closure trigger of 2.13 m, the frequency for the present day is 0.20 year$^{-1}$ and it will trap 7.2 million m$^3$ year$^{-1}$ river water (assuming gates close/open at slack tide). With 0.3/0.6/1 m SLR, the frequency increases to 0.36/3.2/42.7 year$^{-1}$ and the annual trapped river water volume increases to 11/76/1240 million m$^3$ respectively. This exponential increase of the annual trapped river water volume by SLR would cause problems with poor circulation/flushing of the estuary, and such a high closure frequency would likely not be permissible.

If the gates close at low tide instead of slack tide (Figure S6 in Supplementary Materials), the river water behind the barrier will begin to be trapped from a relatively lower water level compared with gates closing at slack tide, and this will reduce the probability of trapped river water flooding. However, the gate closure operation at low tide when the water velocity is high is more challenging. This would also increase the gate closure duration, which would increase the volume of water trapped by gate closures. Both strategies are tested in the trapped river water flood risk analysis and results indicate that there is only a very small probability (below 0.005 year$^{-1}$) of trapped river flood for both strategies under 0.6 m SLR based on the extreme value analysis with a generalized Pareto distribution.

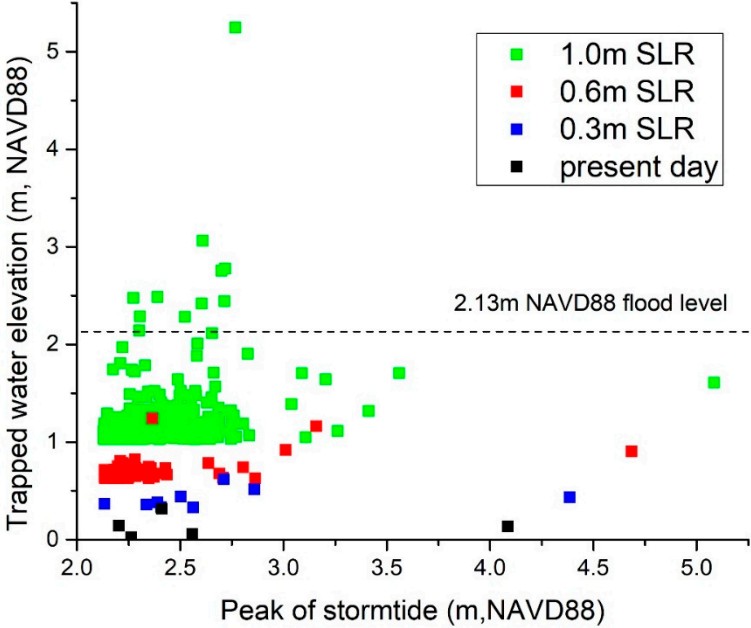

**Figure 9.** Trapped river flooding analysis results based on constant 2.13 m flood trigger assuming gates close/open at slack tide (almost mid tide at the three barrier locations). The scatter points are the historical gate closure events (black) or future gate closure events with 0.3 m, 0.6 m and 1 m SLR (blue, red, green). The horizontal axis shows the peak water level of the storm tide events; the vertical axis is the corresponding trapped water elevation at the Battery location. The horizontal line is the 2.13 m flood threshold at the Battery gauge station.

## 4. Discussion

Our results illustrate that SLR will cause an exponential increase in the gate closure frequency and also an increased duration, if a constant water-level trigger is used to manage gate closures (Figures 4–6 and 8). This will cause the arrival of the USACE's maximum allowable gate closure frequency in the near future. Based on our constant gate closure AEP results (Figure 7), if using the 2.13 m flood trigger, the annual closure frequency will exceed 0.5 year$^{-1}$ around 2060 based on RCP

central estimate of sea level rise results. Therefore, for a 100-year planning horizon, the constant-trigger AEP management regime is needed, and seawalls would need to be raised higher for protection against more frequent floods.

Some of the lower waterfronts in the region have no seawalls and already experience monthly flooding (e.g., Hamilton Beach, bordering Jamaica Bay) or have areas with seawalls as low as 1.7–1.8 m NAVD88 elevation (e.g., Coney Island and Belle Harbor, [31]; and Hoboken). Obviously, a surge barrier with a trigger water level of 2.13 m will not help these neighborhoods avoid frequent flooding, also known as "residual risk". To prevent flooding for these locations, seawalls will immediately be needed to be built or raised. At a typical cost of $20 million per kilometer of length and meter of elevation for seawalls/levees [32], and on the order of 100 km of waterfront likely requiring new construction to reach the 2.13 m protection level, this will add billions of dollars to any comprehensive plan that seeks to address both surge and SLR. Most likely, costs and benefits will need to be weighed on a neighborhood-by-neighborhood basis, and in some cases retreat may be a better option.

If seawalls heights are raised uniformly to 2.13 m, the combined surge barrier/seawall system's useful time horizon (UTH) could be as short as 2040, if accounting for high-end sea level rise and forecast uncertainty (Figure 7). Alternatively, the UTH could be 2090 if using a central estimate of sea level rise and no forecast uncertainty (Figure S2 in Supplementary Materials). Lastly, the UTH could extend into the 22nd century if sea-level rise trends lower than the central estimate.

Wave-driven overtopping of seawalls makes the problem of sea-level rise even more challenging, and was not quantified in this study. New York Harbor significant wave heights during a typical 0.5 year$^{-1}$ AEP storm tide are approximately 1 m in height, with crests 0.5 m above the still water-level. At downwind locations with a large enough wind fetch, frequent overtopping will occur if the waterfront height is only high enough to stop the still-water level. To comprehensively assess seawall heights and costs, a study would need to include wave modeling and consider the allowable overtopping volumes.

### 4.1. Sensitivity to Uncertainty in SLR

There is a large uncertainty for the gate closure frequency (Figures 4–6) and the barrier/seawall useful time horizon (Figure 7) in the future due to the high uncertainty of SLR. If using the central estimate of SLR to plan for the gate closure trigger and required shoreline elevation, the sea level could rise more rapidly, in which case the shoreline elevation would need to be further raised to adapt to a higher SLR. Planning for the 90th percentile SLR trajectory up front will substantially increase the construction cost of the barrier/seawall system, but this can avoid the risk of requiring future seawall reconstruction. Even if SLR does not occur at such a high rate, the high waterfront elevation can extend the useful lifetime of the barrier/seawall system.

Applying USACE SLR projections (2020–2100) for the constant frequency analysis and comparing it with other SLR projections shows that USACE's intermediate SLR is below the central estimate and well below the 90% percentile of either RCP4.5 or RCP8.5 (Figure 7). The USACE intermediate SLR projection is used by USACE for its cost-benefit analysis ([1], p. 18 in Economics Appendix). Therefore, the Harbor and Tributaries Focus Area Feasibility Study (HATS) may underestimate the cost of seawalls and the benefit of reduced flood damage in their cost-benefit analysis.

### 4.2. Sensitivity to Forecast Uncertainty

In the practical operation of the surge barrier system, gates must be closed if there is a reasonable likelihood (e.g., 5%) of exceeding the trigger water level, for risk-averse purposes. The incorporation of forecast uncertainty based on data from an existing operational ensemble flood model shows that it leads to an increase in closure frequency. This requires a higher trigger water level, in order to keep closures at the acceptable frequency of 0.5 year$^{-1}$ (Figure 7 v.v. Figure S2 in Supplementary Materials). This impact should be taken into consideration for the gate closure frequency prediction and benefit-cost analysis.



In the future, forecast precision could be improved as weather and ocean modeling techniques improve. However, there will always remain aleatoric uncertainty due to natural processes, even if models were to become perfect. As forecast uncertainties will shrink, uncertainty will always exist, and the actual closure frequency is likely to be between the two cases shown in Figure 6.

### 4.3. Sensitivity to Interannual Variability in Storm Surge and Mean Sea Level (MSL)

The gate closure frequency is only sensitive to storm surge and MSL interannual variability in the first 3–4 decades when SLR uncertainty is low (Figure S3 in Supplementary Materials). This analysis requires a more complex analysis approach for synthesizing future water level exceedance data. We use a Monte Carlo approach to merge these sources of interannual variability with SLR uncertainty by superimposing 20,000 samples of SLR on random past years of storm tide data. In this way, we can convolve the empirical annual flood frequency distribution together with the probabilistic SLR distributions, from which we obtain the distribution of single-year closure frequency at each at decadal time step, instead of an annual-average closure frequency. We use the 90th percentile from the distributions to represent the "single-year max annual closure frequency". With the large uncertainty of SLR beyond 30 years into the future, results become insensitive to interannual variability.

### 4.4. Low Near-Term Risk of Trapped River Water Floods

If a constant trigger water level is used, SLR will increase the gate closure duration and frequency (Figure 8) which will increase the risk of trapped river water flooding. This is shown using historical events in Figure 9. However, the probability of trapped river water flooding for cases of SLR of 0.6 m and below is still below 0.005 year$^{-1}$ for either the gates closed at the low tide or slack tide. Our trapped water analysis results are conservative (i.e., if anything, biased high). First, some types of smaller auxiliary flow gates (not the large navigational gates) may have the capability to be partially opened to release some water when the offshore water level is periodically near low tide (cases shown in Figure 2). If gates have this capability, then this could be done anytime the offshore water level dips below the inshore water level. In addition, the hydrodynamic model we use does not have floodplains, and hence the trapped water level could have a high-bias due to not being able to spread into a floodplain. Nevertheless, the results still show a low risk of trapped river water flooding for up to 0.6 m SLR.

Looking further into the future, a higher SLR scenario of 1 m results in more risk of trapped water floods exceeding the 2.13 m flood threshold (Figure 9). Here, the future barrier management regime becomes important, because the annual number of closures for sea-level rise of 0.6 m is more than 3 year$^{-1}$, and for 1.0 m is above 42 year$^{-1}$, far exceeding the rate which the USACE has said would be acceptable. Thus, the frequency of closure would likely be unacceptable well in advance of the arrival of this quantity of SLR. If the management is based on a constant AEP trigger, then trapped water flooding will always remain a very low-probability event.

Considering that future rain intensity could be higher than the present-day rain intensity, due to climate change, there could be an increase in streamflow. Based on climate change research for the New York area, the runoff volume increase is less than 30% for the 2-year/50-year extreme events [33]. Similarly, another study found that the rain intensity increase is less than 22% for extreme events [34]. However, the worst of our historically-based trapped river flood events, even with 0.6 m SLR (Figure 9), would need to increase by about 80% to reach the 2.13 m flood level at the Battery. While we neglect the effect of future rainfall increases in our analysis, these studies suggest this effect would not significantly alter our conclusions above.

A closer look at the trapped water modeling results with no floodplain (Figure S5 in Supplementary Materials) reveal a fairly linear relationship between trapped water level and volume, where the slope is related to the tidal waterway area. This suggests a simplified method of quantifying trapped river water levels without hydrodynamic modeling, which is to assume there is no floodplain and compute water level as time-integrated water volume divided by the area of tidal waterways behind the barrier. The assumption of no floodplain is reasonable if there is high topography, as is the case in most of the

Hudson River valley, or if seawalls are blocking the flooding, which is correct up until the point where flood stage exceedances occur (which is the point we are interested in capturing). One must provide a gate closure duration and estimate the total freshwater input to the system, which can often be different from the total that is measured at river gauges. This can be done by scaling up the measured volume by the ratio of total (ungauged plus gauged) watershed area divided by the gauged watershed area [35].

## 5. Conclusions

This study quantifies how SLR would influence the closure frequency, duration, and trapped river flood risk for a storm surge barrier system being considered for the New York City metropolitan area. The research demonstrates a transferable framework that includes a combination of historical observations, superposition of sea-level rise, and optional computational hydrodynamic modeling. The results indicate the trigger water level and the forecast uncertainty both strongly influence the annual number of gate closures. With gate closure management based on a constant trigger water level, SLR causes an exponential increase of the gate closure frequency and a lengthening of the closure duration. With gate closure management based on a constant AEP, a useful time horizon is defined for the barrier/seawalls system, which help inform the required seawall heights and the HATS benefit–cost analyses. If seawalls heights are raised uniformly to 2.13 m, the system's useful time horizon could be as soon as 2040, accounting for high-end sea-level rise and forecast uncertainty, or 2090 with a central estimate of sea-level rise and no forecast uncertainty, or longer if sea-level rise trends are lower than the central estimate. The probability of trapped river water flooding is presently very low, but increasing sea levels lead to an increase in this probability. However, closure frequency rises well above 0.5 year$^{-1}$ before this becomes a problem and hence closure frequency is a more immediate problem that could be addressed by raising the trigger water level.

**Supplementary Materials:** The following are available online at http://www.mdpi.com/2077-1312/8/9/725/s1. Figure S1: Scatterplot is the top 20 high-water level events showing the storm surge and its corresponding "high-end uncertainty" 24 hours before the event (from an ensemble forecast) in the past 4 years. The red line is the "forecast uncertainty" regression model with the 95% prediction interval (black dash line). Figure S2. Water level trigger future evolution by allowing constant AEP (0.5 year-1) based on RCP 4.5 (black) and RCP 8.5 (blue) future greenhouse gas emissions trajectories and the USACE intermediate SLR projection (red); these results are using observed data only (no forecast uncertainty has been incorporated), in contrast to Figure 7. Figure S3. Gate closure frequency analysis results based on constant 2.13 m flood trigger and RCP 4.5 future greenhouse gas emissions trajectory; Black dashed line is the 90th percentile single-year closure frequency. Dashed line is as Figure 4. Figure S4. The study area with the USGS gauge locations (Purple dots), the three barriers' closure system (red dots) and the Battery tide gauge location (Green square). Figure S5. The relationship between water elevation rise at Battery and total accumulated water volume. Different colors are corresponding to different historical flood events with multiple points for different time of the event; the points recorded the modeled water rise at Battery and the corresponding total freshwater input; the dash lines are the linear fit of each flood event; the black solid line is the final relationship fitted by using all the points. Figure S6. Trapped river flooding analysis results based on constant 2.13 m flood trigger assuming gates close/open at low tide, in contrast to Figure 9.

**Author Contributions:** Conceptualization, P.O., Z.C.; methodology, P.O., Z.C., T.W.; software, Z.C.; validation, Z.C., P.O.; formal analysis, Z.C., P.O.; investigation, P.O., Z.C.; resources, P.O.; data curation, Z.C., P.O.; writing—original draft preparation, Z.C., P.O.; writing—review and editing, P.O., Z.C., T.W.; visualization, Z.C., P.O.; supervision, P.O., T.W.; project administration, P.O.; funding acquisition, P.O. All authors have read and agreed to the published version of the manuscript.

**Funding:** This research was funded by the NOAA National Estuarine Research Reserve Science Collaborative (award NA14NOS4190145), the New York State Energy Research and Development Authority (NYSERDA; agreement 145583) and the National Oceanic and Atmospheric Administration (NOAA) Climate Program Office (CPO) Regional Integrated Sciences and Assessments program (RISA; award NA15OAR4310147).

**Acknowledgments:** The authors are grateful to Stefan Talke (Cal Poly) for providing water level data used in some analyses, and project collaborators Bennett Brooks (Consensus Building Institute), Kristin Marcell (New York State Department of Environmental Conservation, NYSDEC) and Sarah Fernald (NOAA Hudson River National Estuarine Research Reserve and NYSDEC). Thanks also to the USACE New York District for assistance in the research.

**Conflicts of Interest:** The authors declare no conflict of interest.

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
