# Peer review of "Storm Surge Barrier Protection in an Era of Accelerating Sea-Level Rise: Quantifying Closure Frequency, Duration and Trapped River Flooding"

_jmse, doi:10.3390/jmse8090725_

Round 1

Reviewer 1 Report

Thank you for clariying all my questions.

Some aspects have also been improved.

Reviewer 2 Report

The authors addressed most of my concerns adequately. I recommend accepting this paper

This manuscript is a resubmission of an earlier submission. The following is a list of the peer review reports and author responses from that submission.

Round 1

Reviewer 1 Report

This is an interesting paper. The authors use a simplified approach to study the quantification of closure frequency and duration for storm surge barriers under sea-level rise conditions. The results will be useful for shoreline management and protection. The paper is well written overall, but the authors do not explain the approach well and it is difficult for readers to repeat the analysis. I have some concerns and would like the authors to address them.

  1. The storm surge barrier is used for the protection of inundation during storm surge events. The analysis focuses on the normal and minor storms, rather than storm surge. If the surge barrier needs more frequent closure, it implies that the current shoreline is not protective. It appears that the analysis results show that the current lower waterfronts areas are no longer protective. My confusion is that the study results do not relate to storm surge, rather the results are indicative that water level will frequently above the current flood trigger level under sea-level rise condition. This analysis can be used anywhere without a storm surge barrier. I am not sure this frequency analysis is useful for a storm surge barrier? To me, the frequency analysis does not need to be for a storm surge barrier. The approach is useful for any estuaries and harbors.  
  2. It will be good to provide a diagram showing mean lower low water, mean sea-level, mean higher high water, 2.13 m flood trigger, and flood triggers under different sea-level conditions, which can provide an overall picture of the current status of the region.
  3. The authors stated that “we create a semidiurnal maxima dataset from the detrended hourly water level data to represent the stochastic storm and tide driven signal in harbor water levels.” First, if the authors ignore any changes to tides or storm surge, what kind of ‘stochastic storm’ does it referred here? Second, I am not clear how to create the ‘stochastic storm-driven’ and ‘tide-driven variability’ in harbor water levels. If the authors used current measurements to create this data set, how do you use it for the further under sea-level rise for different years? Do you create a stochastic storm-driven model? I suggest the authors providing detailed procedures and steps for the analysis so that the readers can repeat the analysis. If the procedure can be expressed mathematically, it will be clearer.
  4. Line 170 “The SLR projections are stored in 20,000 Monte Carlo samples at decadal time steps” I have no idea what Monte Carlo samples are generated? Do you use harmonic analysis or use the ‘semidiurnal maxima dataset’
  5. When analyzing discharge impact on duration, did the authors considered spatial variations of water level at different locations? The authors did mention that the numerical model results were used for the analysis, but I still not clear if the accumulation of water is referred to at a fixed location. I am not sure how to extend current information of discharge to the further as firewater discharge varies greatly each year.
  6. Figure 6. What are three dashed lines? Please explain them in the caption.  

Reviewer 2 Report

The manuscript presents the quantification of the effect of climate change (SLR) in the frequency and duration of the closures of the surge barrier gates and in the probability of trapped river water flooding for the New York City metropolitan area. The work is innovative because only the gate closure frequency has previously been investigated. A transferable methodology has been developed combining instrumental data (tide gauge), probabilistic SLR projections and modelling data (numerical simulations of the trapped water elevation using a hydrodynamic model). Although several simplifications along the steps of the methodology have been adopted, they have been justified and discussed.

The manuscript is worth to be published but I found several aspects not enough clear and I would like that you clarify them.

The gate closure frequency analysis is provided with confidence intervals. As I understood, the analysis is performed using historical data (1920-2019) of harbor water levels from a tide gauge located at Battery and the uncertainty is due to SLR projections (the 10th, 50th and 90th percentile of SLR). Therefore, does the 80% confidence interval represents the results of the 10th and 90th percentile of SLR? Or have you applied a probabilistic model for the storm surge?

Regarding the method you have applied to introduce the forecast uncertainty of water level, you mentioned in the text (lines 142-144) that the uncertainty is usually larger for more extreme events but in the scatter plot showing the storm surge and its corresponding “high-end uncertainty” 24 hours before the event (Figure S1), the uncertainty for the most extreme events seems to be underestimated. As I understood, the forecast uncertainty is just a certain value, which depends only on the surge, added to historical observations but does not introduce uncertainty that could be quantify and express as confidence intervals in the final results of the gate closure frequency. Could you apply a more sophisticated model or work only with the most extreme events to improve this relationship and the concept of uncertainty?

The trapped River Flooding Analysis is the most interesting part of your work. Although you cited several previous references where the hydrodynamic model has been applied in the domain area, a figure that shows the study area with the USGS gauge locations (16 in this study, 9 gauges in other references), the three barriers’ closure system, the Battery location could help to understand how the system works (I found difficult to give myself an idea of it). I wonder if a figure with one of the simulations could also help. It is also not clear for me what “the peak of the water level of the flood events” in Figure 8 and Figure S5 represents. Is this value used in the analysis?

Why was the likelihood of future flooding from trapped river water only performed for future SLR scenario/decade when the number of empirical cases of gate closure is greater than or equal to 25 times? Why did you fit the GPD only to the top 25 trapped river flood elevation cases?